# An Insight into the Sorption Behavior of 2,3,7,8-Tetrachlorodibenzothiophene on the Sediments and Paddy Soil from Chaohu Lake Basin

**DOI:** 10.3390/ijerph191811346

**Published:** 2022-09-09

**Authors:** Kainan Nian, Wenli Xiong, Yalu Tao, Ziqing Zhu, Xiaoxue Pan, Kang Zhang, Xuesheng Zhang

**Affiliations:** 1School of Resources and Environmental Engineering, Anhui University, Hefei 230601, China; 2Laboratory of Wetland Protection and Ecological Restoration, Anhui University, Hefei 230601, China; 3China Science and Technology Intelligent Agricultural Valley Collaborative Innovation Research Institute, Hefei 231131, China

**Keywords:** 2,3,7,8-TCDT, Freundlich isotherm, dissolved organic matter, organic carbon normalized partitioning coefficient, Chaohu Lake

## Abstract

Considering the frequent detection of polychlorinated dibenzothiophenes (PCDTs) in various environmental matrices and the potential ecological health risks, the environmental behavior of such compounds needs to be elucidated further. In this work, the sorption behavior of 2,3,7,8-tetrachlorodibenzothiophene (2,3,7,8-TCDT) onto three sediments and paddy soil from Chaohu Lake were investigated via batch equilibration experiments. From the perspective of sorption kinetics and isotherms, the sorption characteristics and mechanism of 2,3,7,8-TCDT on the above four carriers were compared, and the relationship between their structural characteristics and soil sorption capacity was discussed. Results suggested that rapid sorption played the primary role during the sorption process of 2,3,7,8-TCDT and the corresponding sorption isotherms were well fitted using the Freundlich logarithmic model. Moreover, the effects of pH and dissolved organic matter (DOM) on the sorption of 2,3,7,8-TCDT were investigated. The maximum sorption capacity of 2,3,7,8-TCDT on sediment was under acidic pH condition (pH = 4.0). Meanwhile, DOM at a low level promoted the sorption capacity of sediment toward 2,3,7,8-TCDT, while the high concentration of DOM inhibited this effect. In addition, the values of log*K*_oc_ were obtained using high-performance liquid chromatography (HPLC) and did not show any significant correlation with organic carbon (OC) contents, thereby indicating that the partition effect was the dominating influencing factor for the sorption of 2,3,7,8-TCDT both on sediments and soil. This work provides useful data to understand the sorption behavior of 2,3,7,8-TCDT on sediments and soil and assess its potential environmental risk.

## 1. Introduction

Polychlorinated dibenzothiophenes (PCDTs) structurally resemble polychlorinated dibenzofurans (PCDFs) and are considered as sulfur-containing dioxin-like compounds [1] (Figure 1). In terms of the substitution numbers and sites of chlorine atoms, PCDTs include 135 congeners [2]. The major sources of PCDTs in the environment are the incineration of municipal and hazardous wastes, the high-temperature process of metallurgy and metal recovery, and the manufacturing process of polychlorinated biphenyls (PCBs) and trichlorobenzene sulfonate [3,4].

Owing to the above anthropogenic sources, PCDTs have been detected in many environmental samples such as fly ash, soil, sediments, and aquatic organisms [5,6,7,8,9] (Appendix A). It has been reported that the levels of PCDTs in fly ash were 55 ng·g^−1^ in incineration samples and the level of 2,3,7,8-tetrachlorodibenzothiophene (2,3,7,8-TCDT) was 35 ng·g^−1^ [6]. The levels of 2,3,7,8-TCDT in crabs and lobsters from the United States were 8.3 and 1.0 ng·g^−1^, respectively [7]. Furthermore, PCDT homologs in some samples are observed at higher levels than the corresponding polychlorinated dibenzo-dioxins/furans (PCDD/DFs). It has been found that the concentration of 2,4,6,8-tetrachlorodibenzothiophene (2,4,6,8-TCDT) was approximately 5.5 times as high as that of 2,3,7,8-tetrachlorodibenzodioxin (2,3,7,8-TCDD) in the sediment samples of the Passaic River (NJ, USA) [10]. And their relatively specific values were 3680 and 656 pg·g^−1^. In addition, due to the similar structures of PCDTs and PCDD/Fs, the ecotoxicity of PCDTs has aroused much attention from researchers worldwide. The results showed that mouse hepatoma cells exposed to PCDTs could cause the induction of aryl hydrocarbon hydroxylase and ethoxyresorufin-*o*-deethylase (EROD) activity [11,12]. Blue crabs (*Callinectes sapidus*) collected from Newark Bay, at the mouth of the Passaic River, contained higher contents of 2,4,6,8-TCDT than that of 2,3,7,8-TCDD in their muscle and hepatopancreatic tissues [7]. There will be tissues bioaccumulation and hepatic oxidative damage in *Carassius auratus* exposed to 2,3,7,8-TCDT [13]. Obviously, PCDTs have higher potential environmental risks and ecological health risks than PCDD/DFs do, and it is necessary to clarify the environmental behavior of PCDTs.

Sorption is a very important environmental behavior for organic compounds, especially for those with high log*K*_ow_ values, such as polychlorinated diphenyl ethers (PCDEs) and PCDTs. This process plays a key role in the migration, transformation, and bioavailability of contaminants [14]. Previous studies have found that sorption is affected by many environmental factors. For instance, rapid sorption played a primary role during the sorption process of the selected PCDEs in soils and their sorption isotherms were fitted using the Freundlich logarithmic model [15]. The sorption results of dimethyl phthalate (DMP) and diethyl phthalate (DEP) on two soils showed that the soil organic carbon (OC) content played a major role in the sorption and pH could affect the adsorption capacity [16]. The nature and content of both sediment-bound and dissolved organic matter (DOM) dominate the sorption of hydrophobic organic contaminants in the estuary [17]. Additionally, pH value, ionic strength, and DOM can affect the adsorption–desorption behavior of triclosan in the water–sediment system [18]. Moreover, the sorption-desorption of perfluorooctanoic acid (PFOA) onto ten soil samples depended on soil OC content and the composition of soil minerals [19]. It is consequently hypothesized that these reported factors may also affect the sorption behaviors of PCDTs on sediments and soil, and their specific effects and the underlying sorption mechanisms should be systematically studied.

As one of the five major freshwater lakes, Chaohu Lake plays a crucial role in the agricultural development of Anhui province, China. Hefei city is located on the shore of the west lake of Chaohu Lake, while other shores are surrounded by vast areas of fertile farmland [20]. Eight major inflow rivers can introduce industrial and domestic sewage, and unrestricted surface runoff discharged by agricultural non-point source pollution into Chaohu Lake [21]. Therefore, the pollution of nitrogen, phosphorus, and persistent organic pollutants (POPs) in Chaohu Lake get more attentions by researchers. However, only a few studies have focused on emerging contaminants. Previously, our findings have confirmed the prevalent existence of PCDTs in different matrices of Chaohu Lake [22], and the concentrations of 14 PCDTs in sediment, suspended particulate matter (SPM), and surface water are 0.40–3.55 ng·g^−1^ (dry weight, d.w.), 0.38–2.95 ng·g^−1^ d.w., and 0.34–2.61 ng·L^−1^, respectively. Additionally, it is found that sediments are their source and sink, so it is very important to study the sorption behavior of PCDTs in sediments. Likewise, soil is another important source and sink of PCDTs, and sorption of these compounds in soil can simultaneously determine their fate, transport, and transformation in the environment [23]. Considering that rice is one of the main crops in the study area, paddy soil was also collected as sorption carrier. Thus, not only the sorption of PCDTs in the sediments but also the sorption mechanism between the soil and PCDTs needs to be considered to better evaluate their risk.

In this context, 2,3,7,8-TCDT (similar in structure to 2,3,7,8-TCDF, which is the most toxic among PCDFs) was selected as the target pollutant. Its sorption behavior in sediments from Chaohu Lake and typical paddy soil from Chaohu Basin were investigated in the current study. The specific objectives were (1) to investigate the sorption kinetics and isotherms of 2,3,7,8-TCDT via batch equilibrium experiments; (2) to discuss the effects of different key environmental factors (i.e., soil (sediment) type, DOM, and pH) on the sorption of 2,3,7,8-TCDT; and (3) to measure and compare the organic carbon content normalized sorption coefficient (log*K*_oc_) values of 2,3,7,8-TCDT in different sediment and soil samples.

## 2. Materials and Methods

### 2.1. Chemicals and Reagents

Sodium hexametaphosphate ((NaPO_3_)_6_) was supplied by Tianjin Guangfu Chemical Research Institute (Tianjin, China). Analytical reagent (AR) grade hydrochloric acid (HCl), hydrogen peroxide (H_2_O_2_), mercury chloride (HgCl_2_), sodium chloride (NaCl), and potassium dichromate (K_2_Cr_2_O_7_) were purchased from Sinopharma Chemical Reagent Co., Ltd. (Shanghai, China). 2,3,7,8-TCDT was purchased from Wellington Laboratories (Ontario, Canada). High-performance liquid chromatography (HPLC) grade methanol (CH_3_OH) was supplied by TEDIA company (Ohio, USA).

### 2.2. Sample Collection and Characterization

The surface sediments were sampled from three parts of Chaohu Lake, and the soil was collected from ecological paddy fields (Lujiang, Hefei, Anhui) (Appendix A). As depicted in Appendix A, site S1 was located in the western part of Chaohu Lake with two inflow rivers (e.g., the Nanfei River and Pai River). These tributaries could discharge organic contaminants generated by the production and living activities from Hefei city into Chaohu Lake. Site S2 was located in the middle of Chaohu Lake, a possible sink of industrial wastes, around which there are oil refining and processing enterprises, and S3 was located in the eastern part of Chaohu Lake with less pollution as compared to the western part and middle of Chaohu Lake. Surface sediments (0–12 cm) were collected with a stainless-steel grab sediment sampler, refrigerated at 0–4 °C, and transported back to the laboratory within 12 h. Paddy soils were sampled from ecological integrated planting and rearing model paddy field around Chaohu Lake. Prior to analysis, sediment and soil samples were stored at −20 °C [24,25]. The OC, soil texture, total nitrogen, and pH of the soils were measured according to previously reported soil testing procedures [26]. The physicochemical properties of sediments and soil are shown in Table 1.

### 2.3. Extraction and Preparation of DOM

DOM was extracted from decomposing rice straw according to the water extraction method. The specific procedures were as follows: rice straw was mixed with distilled water in glass bottles, and the above supernatant was centrifuged at 22 °C for 24 h. After filtering through a 0.45 μm sterile membrane, the obtained filtrate was stored at 4 °C. The basic properties of the obtained DOM are represented in Table 2.

### 2.4. Batch Equilibrium Experiment

Batch equilibrium experiments were conducted to study the sorption kinetics and isotherms of 2,3,7,8-TCDT. The initial concentration (C_0_) of 2,3,7,8-TCDT was 1 mg·L^−1^. The 0.01 M CaCl_2_ were used as the aqueous solvent phase which can balance the concentrations of ions in the solution and to improve centrifugation and minimize cation exchange, and 0.05% HgCl_2_ was added as a biocide to inhibit microbial degradation [15]. The ratio of the soil mass (g) to the solution volume (mL) of compounds was 1:100. The experiments were conducted in the dark at a constant room temperature (25.0 ± 1.0 °C) for 72 h in a reciprocal shaker. The solution pH was maintained at 7.00 ± 0.10 with 0.8 mM NaOH and 0.08 mM HCl.

In the kinetic studies, soil suspensions were sampled at prespecified time intervals (0.2, 0.5, 1, 1.5, 2, 3, 7, 12, 24, 48, and 72 h) and centrifuged at 6000 rpm for 15 min. The concentrations of the substrate in the solution were quantified by HPLC. To evaluate the sorption linearity, the full range isotherms were obtained through sorption experiments using different concentrations of the test substances (i.e., diluted 2, 4, 10, or 20 times). An equilibration time of 12 h was chosen based on the preliminary kinetic experiments. After equilibrium, the suspension was centrifuged. The aqueous concentrations of 2,3,7,8-TCDT before and after the sorption process were determined by HPLC.

### 2.5. Influencing Factors of Isothermal Sorption

In order to investigate the effects of different soil types on adsorption, filtered sediments or soil samples, distilled water, 0.01 mol/L CaCl_2_, and 5 mg/L HgCl_2_ were accurately added into 250 mL conical flask in sequence. After injecting the same volume of five different concentration gradients of 2,3,7,8-TCDT stock solutions (in order to avoid the eutectic effect, the volume of methanol content in the solution is not more than 0.5%), the mixtures were put in thermostatic oscillator (25 °C, 180 r/min, 48 h, in dark). After shaking, the above solutions were centrifuged at 6000 r/min for 20 min. The supernatant was determined by HPLC, and the sorption content of 2,3,7,8-TCDT by sediment was calculated by different methods. Each group was performed in triplicates and the blank experiment was conducted at the same time.

In order to determine the effect of pH on 2,3,7,8-TCDT in sediment, we selected sediment samples from Eastern Chaohu Lake. The pH values of the soil solution were adjusted to 4.0, 5.5, 7.0, 8.5, and 10.0 by HCl and NaOH, respectively. Then, the 2,3,7,8-TCDT stock solution was added with different concentration gradients. The following steps were the same as above.

In order to examine the changes in sorption capacity of 2,3,7,8-TCDT under different concentrations of DOM, the sediment samples from the eastern part of Chaohu Lake were selected and added with 0, 1.0, 5.0, 10.0, 50.0, and 100.0 mg/L of the DOM background solution, and the pH value of the solution was kept at 8.0. Moreover, the changes in DOM concentration were measured by the total organic carbon (TOC) analyzer. Other specific operations were as mentioned above.

### 2.6. Determination of logKoc

The log*K*_oc_ values were determined following the established method in the previous study [27]. A certain amount of air-dried sediment or soil samples were accurately weighed and placed in triangular bottles with plugs (polytetrafluoroethylene stoppers), while PCDTs with high hydrophobicity were selected in glass dragon-mouth bottles. To ensure that the sorption percentage was higher than 20% and the ratio of soil to water was lower than 80%, the background solution containing 0.01 mol/L CaCl_2_ and 5 mg/L HgCl_2_ was added. Then, equal volumes of five different concentration gradients of 2,3,7,8-TCDT were added, and the cover was placed in a water bath thermostatic oscillator, which was oscillated at 180 r/min for 48 h at room temperature (25 °C) in the dark. After shaking, the samples were centrifuged at 6000 r/min for 20 min, and the concentrations of PCDTs in the supernatant were directly determined by HPLC or exacted by the liquid-liquid extraction method prior to HPLC. Specifically, for PCDTs with high hydrophobicity, the supernatant should be firstly extracted using *n*-hexane. After being concentrated and solvent rotary evaporated, the samples were re-dissolved in methanol to 1 mL, and then determined by HPLC. The recovery rate was between 85% and 115%.

The calculation formula of *K*_oc_ was as follows [28,29]:*K_d_ = q_e_/C_e_*(1)
*K_oc_ = K_d_/f_oc_*(2)
where *q_e_* was the concentration of the compound in the soil phase at equilibrium mg/kg; *C_e_* was the concentration of the compound in the aqueous phase at equilibrium mg/L and *f*_oc_ represented the soil organic carbon content.

### 2.7. HPLC Analysis

Samples were analyzed in an HPLC system. The chromatographic conditions were as follows: samples were eluted and separated by the C_18_ column (150 mm × 4.6 mm, 5 μm, Agilent Company, Santa Clara, CA, USA); the mobile phase was 80% methanol and 20% water (contains 0.05% trifluoroacetic acid). The flow rate was 1 mL·min^−1^ and the injected volume of the sample was 10 μL. The detection wavelength of 2,3,7,8-TCDT was selected as 242 nm.

### 2.8. Quality Assurance (QA)/Quality Control (QC)

Procedural blanks, method quantification limits, and recovery experiments were used to evaluate the credibility of the analysis method. A procedural blank was run after the analysis of each batch of samples (*n* = 3), and none of the target compounds were detected in these blank samples. The method detection limits (3 times the signal-to-noise ratio) of 2,3,7,8-TCDT in sediments and soil ranged from 0.15–0.45 μg·g^−1^ d.w., and the recoveries of 2,3,7,8-TCDT in different sediments and soil were ranged from 85.2% to 106.9% (Appendix A). The levels of 2,3,7,8-TCDT were calculated from the corresponding calibration curve, which had a correlation coefficient of *R*^2^ = 0.9994. The detailed descriptions of the preparation of the calibration curve can be found in SI. In addition, to assess the sample losses during the experimental procedures, mass balance checks were carried out in the preliminary experiments. The concentrations of 2,3,7,8-TCDT in both the soil and test vessel extracts were determined, and the mass balance recoveries, which were in the range of 85.4% to 103.3% (mean value: 93.7%), were measured. Such results suggested that 2,3,7,8-TCDT is considered to be stable during the entire experimental period.

### 2.9. Data Analyses

All data are shown as the means ± standard deviation (SD) and were analyzed in Origin2017 (OriginLab Corp., Massachusetts, USA). Statistical differences among different groups were analyzed in the SPSS 26.0 software package (SPSS Inc., Chicago, IL, USA) using a one-way analysis of variance (ANOVA) and Dunnett’s test. Values of *p* < 0.05 were considered significant. Figures were prepared using GraphPad Prism 8.0 software (GraphPad Software, California, USA).

## 3. Results

### 3.1. Sorption Kinetics

The sorption kinetics of 2,3,7,8-TCDT on four tested solids are shown in Figure 2. In general, two stages of the sorption process of 2,3,7,8-TCDT on these carriers could be revealed. First, significant sorption occurred at the initial 2 h, with sorption percentages of 46.5%, 54.1%, 66.1% and 61.7%. Then, the sorption gradually increased at a much slower rate and eventually saturated after nearly 12 h for all of the samples, indicating that the dynamic equilibrium was reached. At this time, the sorption percentages of 2,3,7,8-TCDT in 12 h were 57.0%, 63.1%, 71.4%, and 69.0%, respectively. As shown in Figure 2, for all of the four samples, the maximum sorption capacity followed the order from high to low: Western part of Chaohu lake > Paddy soil sample > Middle of Chaohu Lake > Eastern part of Chaohu Lake, which was consistent with the order of their soil sand contents, while negatively correlated with silt contents. The sorption of 2,3,7,8-TCDT in 2 h is accounted for 78.8%, 84.0%, 90.3% and 88.0% of the final sorption. These results suggested that fast sorption plays a primary role in the sorption process of 2,3,7,8-TCDT.

The sorption process of 2,3,7,8-TCDT in the three sediments and paddy soil showed a two-stage characteristic, which could be attributed to the different sorption properties of 2,3,7,8-TCDT on the “rubber” and “glass” organic matter. It is considered that “rubbery” organic matter exists in the outer layer of particulate matter, with loose structure, easy access, fast sorption rate, and linear sorption. While “glassy” organic matter is deeply buried in the interior of particulate matter, with tight structure and difficult to be accessible, and the sorption rate is very slow and present in nonlinear pattern [30]. The fast sorption can be caused by the sorption of 2,3,7,8-TCDT on the easy adsorption sites of the soil surface, and with the sorption progresses, these places were decreased. Thus, the sorption process gradually transformed into the diffusion to the soil internal organic matter, and further resulting in a decrease in the adsorption rates [31]. Meanwhile, the oscillations disturbed the surface structure of soil particles and exposed new adsorption sites for 2,3,7,8-TCDT; therefore, the slow adsorption process still needed a long time to reach the final equilibrium. Slow adsorption may be also caused by the diffusion of organic matter in soil glume micropores and organic matter [15]. Other studies suggested that the rapid adsorption at the initial stage was related to the weak binding site, while the subsequent slow adsorption was caused by the slow allocation of organic pollutants to the strong binding site [32]. In the study of the sorption kinetics of PCDEs in natural soil, it has been found that fast sorption was primary throughout the entire sorption process and contributed mainly to the increase in the total sorption amount in the early sorption stage. On the other hand, the proportion of the slow sorption gradually increased and approached a stable stage at the apparent equilibrium [33]. Therefore, it can be inferred that the sorption kinetic characteristics of 2,3,7,8-TCDT in the three sediments and paddy soils in this study are mainly related to the number and the strength of adsorbed sites.

### 3.2. Sorption isotherms

The sorption mechanism of organic pollutants in soil can be divided into the partition of organic pollutants in soil organic matter and soil sorption [34]. The partition is the dissolution of organic matter to organic substance and the sorption isotherm is linear. Sorption is the surface sorption of soil minerals on organic matters and the sorption isotherm is nonlinear. Therefore, scholars at home and abroad have proposed a number of theoretical models, including the linear sorption model, Langmuir sorption isotherm, and Freundlich sorption isotherm [35]. Previous research results showed that the isothermal sorption behavior of polychlorinated diphenyl sulfides (PCDPSs) in soil conforms to the Freundlich sorption equation [27]. Therefore, the Freundlich sorption model was used to fit the sorption behavior of 2,3,7,8-TCDT in sediments and soil. The sorption isotherms of 2,3,7,8-TCDT are shown in Figure 3, and the Freundlich parameters are listed in Table 3. All the obtained correlation coefficients (*R*^2^) were higher than 0.97, suggesting that fitting results to the Freundlich model were satisfactory.

The *n* value is the correction exponent that designates the nonlinearity of sorption isotherm. The high *n* values (>1.0) indicated slightly nonlinear sorption of the substrate on four selected carriers. When *n* = 1, the Freundlich equation can be simplified into a linear relation, which tends to linear sorption characteristics. It is obvious that the *n* values of 2,3,7,8-TCDT in four samples increase with the increasing soil organic matter. For paddy soil and western part of Chaohu Lake sediment, the *n* values are close to 1, which suggested that the partition was the main sorption mechanism. However, for the eastern part of Chaohu Lake sediment, the *n* values are higher than other samples, indicating the slight non-linearity of 2,3,7,8-TCDT sorption in this sediment. The difference in nonlinear characteristics may be related to the higher organic matter in the sediment of the eastern Chaohu Lake. As discussed above, the “glassy” OM shows the nonlinear process toward the sorption of organic pollutants. Furthermore, its content is usually increased in the high level of soil organic matter [36], finally resulting in the above phenomenon. In addition, it has been reported that the surface sorption of soil clay minerals may also lead to nonlinear sorption [37]. However, in general, although the three kinds of sediment samples and paddy soil samples all have certain nonlinearity, partition was still the main sorption mechanism of 2,3,7,8-TCDT in soil.

The Freundlich parameter *K*_f_ can reflect the adsorption strength. The higher its value is, the stronger the adsorption capacity of the soil to the substance can be obtained. As can be seen from Table 3, the *K*_f_ value of 2,3,7,8-TCDT in western sediment was the highest among four solid samples, accompanied by the highest OC content, suggesting that the western sediment presented the highest sorption performance toward 2,3,7,8-TCDT. A positive correlation between OC content and sorption was also found in previous studies [38]. It greatly affects the sorption of hydrophobic organic compounds in soils [39]. In our study, the organic carbon content normalized sorption coefficients *K*_f-oc_ (*K*_f-oc_ = *K*_f_/*f*_oc_) were thus calculated according to the related experimental data [40]. The *K*_f-oc_ values of 2,3,7,8-TCDT differed depending on the sample type, and they were 43.8, 78.5, 33.5, and 32.0 in the paddy soil, sediments from western, middle, and eastern part of Chaohu Lake, respectively. However, the higher *K*_f_ value of sediment in the eastern part of Chaohu Lake with lower *K*_f-oc_ than those in the middle of Chaohu Lake demonstrated the other existed influencing factors on sorption [41]. Soil organic matter type and soil mineral structure may also contribute to the sorption process [42]. As a result of aging and diagenesis, the soil organic matters become heterogeneous, leading to various sorption affinities [43]. In addition, the clay content of middle Chaohu Lake (11.4%) was higher than eastern Chaohu Lake (11.0%), which was consistent with the value of *K*_f-oc_. It has been reported that there is a positive correlation between clay content and *K*_f-oc_ value of phthalic acid esters (PAEs) [14]. Thus, we may conclude that the clay content in sediments and soil might be also another influencing factor for 2,3,7,8-TCDT sorption.

### 3.3. Effect of pH on Sediment Sorption of PCDTs

As a basic physicochemical index of sediment, pH is the main factor in the sorption of 2,3,7,8-TCDT. In order to investigate the contribution of pH-dependent mechanisms to the 2,3,7,8-TCDT adsorption, we selected one typical sediment (the sediment of eastern Chaohu Lake) whose *f*_oc_ value was the greatest. The experiments with pH modification were conducted and the results described using the Freundlich model are summarized in Figure 4 and Table 4. It can be seen that the nonlinear change in sorption of 2,3,7,8-TCDT was not significant with the changed solution pH value. Under the five pH conditions, the *n* value did not change significantly, but the adsorption capacity of 2,3,7,8-TCDT largely varied. At pH 4.00, 5.50, 7.00, 8.50 and 10.00, the corresponding adsorption constants *K*_f_ values of 2,3,7,8-TCDT were 5.05, 5.01, 3.43, 2.69 and 2.86, respectively. Similar to other typical hydrophobic organic compounds (e.g., PAEs and tetracycline), the *K*_f_ value decreased with the increase of pH, and the maximum adsorption capacity of *K*_f_ in sediments occurred under acidic pH condition (pH = 4.0) [14,44].

Furthermore, the sorption capacity of 2,3,7,8-TCDT varied in different pH ranges. The largest sorption capacity discrepancy was found between pH 4.00 and 7.00, and the relative *K*_f_ values decreased from 5.05 to 3.43, with a 32.1% reduction. The sorption capacity of 2,3,7,8-TCDT also varied within 15.0% at two acidic pH conditions (pH = 4.00 and 5.50), while no significant changes were recorded between pH 8.50 to pH 10.00. Such phenomenon indicated that the significant change of sediment sorption affinity for 2,3,7,8-TCDT was mainly concentrated in the changing process from an acidic environment to an alkaline environment. Previous studies have reported similar results on the sorption behavior of organic matter in sediment [45], which may be due to the release of part of dissolved organic matter in sediment into water (from acidity condition to alkalinity condition), increasing the solubility of organic compounds in water, and thus reducing the amount of soil adsorption. It was found that the increase in solution pH can increase the ionization extent of the SOM functional group, then resulting in the poor affinity of sediments for hydrophobic organic compounds [46]. In addition, it was proposed that a higher pH value makes sediment particles more dispersed, and more sediment particles are not in a discrete state with sediment organic matter, which affects the stability of sediment colloid [47].

### 3.4. Effect of DOM on the Sediment Sorption of PCDTs

The behavior of DOM in sediment–water system was shown to affect the migration and transfer of organic contaminants [48]. In addition, DOM can bind to sediment components as well as organic compounds in solution. Therefore, the sorption of organic contaminants in sediment is also affected by DOM. The effect of additive DOM on the sorption of 2,3,7,8-TCDT on the sediments of eastern Chaohu Lake at pH 8.00 and the parameters of Freundlich sorption isotherm are displayed in Figure 5 and Table 5, respectively. It was observed that the presence of DOM could impact the sorption behaviors of 2,3,7,8-TCDT on sediment and the influence was different with the concentration of DOM. Compared with the solution without DOM, the *K*_f_ values were increased for 5.00 and 10.0 mg/L of DOM additions (92.5% and 43.1% of the increase, respectively), and the sorption capacities of 2,3,7,8-TCDT in the sediment were consequently improved. However, this promoted influence was interrupted and a significantly inhibitory effect occurred when DOM concentrations were 50.0 and 100 mg/L. In the studied DOM levels, the most obvious promoting effect on the sorption behavior of 2,3,7,8-TCDT in sediment was 5.00 mg/L, while the most obvious inhibition was 100 mg/L. In conclusion, with the increase of DOM concentration, the sorption efficiency of DOM on the 2,3,7,8-TCDT shows a trend of first increasing and then decreasing.

To summarize, sediment sorption of 2,3,7,8-TCDT in the presence of DOM is affected by the interaction among DOM, compound, and sediment through a series of complicated processes such as competition sorption, co-sorption, and cumulative sorption. On the one hand, the DOM can bind with organic compounds to add their apparent water solubility, which inhibits the sorption of 2,3,7,8-TCDT in the sediment. On the other hand, the DOM can enter the sediment as organic matter supplements, and the DOM–compound complex can be adsorbed on sediment, which can facilitate the sediment sorption of organic contaminants.

### 3.5. Soil (Sediment) Organic Carbon Content-Normalized Sorption Coefficients

The soil (sediment) organic carbon content-normalized sorption coefficient (log*K*_oc_) of 2,3,7,8-TCDT in different sediments and paddy soil were also calculated (Table 6). Results showed that the log*K*_oc_ values of 2,3,7,8-TCDT in paddy soil samples and sediments from Chaohu Lake were varied with each other, which were in the ranges of 3.58–3.90. The minimal log*K*_oc_ value of 2,3,7,8-TCDT was observed in the sediments from middle Chaohu Lake, while the maxima log*K*_oc_ value of 2,3,7,8-TCDT was found in the sediments from western Chaohu Lake. In general, compounds with a larger log*K*_oc_ could be absorbed onto sediment or soil particles more easily [49,50,51]; therefore, the absorption capacity of 2,3,7,8-TCDT in sediments from western Chaohu Lake was the biggest among four samples. Our recent study found that the log*K*_oc_ of tetrachlorined diphenyl sulfides (tetra-CDPS) congeners, whose structures resemble 2,3,7,8-TCDT, in the sediment–water system in Chaohu Lake ranged from 5.22 to 5.39 (mean value: 5.30), which were relatively higher than those of 2,3,7,8-TCDT [52]. Such results indicated that 2,3,7,8-TCDT might be more easily dispersed in the water phase than tetra-CDPSs. Interestingly, no significant correlation was observed between log*K*_oc_ values and *f_oc_* (*R*^2^ = 0.0992, *n* = 4) (Appendix A), indicating that the partition was the dominating influencing factor of the sorption of 2,3,7,8-TCDT both in sediments and soil [15].

Our previous study showed that the range of log*K*_oc_ values of 2,3,7,8-TCDT in the SPM–water system of Chaohu Lake was 1.93~2.35 (mean value: 2.18) [22], which were obviously lower than that reported in this work. It might be due to the partitioning behavior of 2,3,7,8-TCDT in the real water–SPM–sediment system, which can be influenced by various factors (e.g., degradation of PCDTs in sediments via chemical or biological processes), as well as the sorption may be not achieved its dynamic equilibrium status. In addition, the theoretical log*K*_oc_ value of 2,3,7,8-TCDT was 4.87 calculated by ECOSAR 2.0, relatively higher than the experimental log*K*_oc_ obtained in the current study. Such difference indicated that even under laboratory conditions, a variety of environmental factors could affect the partition of PCDTs in the sediment (soil)–water system. Thus, the detailed sorption mechanism of 2,3,7,8-TCDT in the real aquatic environment should be elucidated in future studies.

## 4. Conclusions

In conclusion, the sorption behavior of 2,3,7,8-TCDT in sediments from Chaohu Lake and paddy soil were investigated and compared for the first time. Our findings suggested that rapid sorption played the primary role in the sorption process of 2,3,7,8-TCDT both in sediments and soil, and its sorption isotherms fitted the Freundlich model well. The ranges of adsorption parameters (*n*) for 2,3,7,8-TCDT are obviously different among different sediments and soil samples, and the *n* in the eastern Chaohu Lake is over 1.1, indicating high nonlinear characteristics. Both pH and DOM contents have significant effects on the sorption behavior of 2,3,7,8-TCDT on sediment and soil from Chaohu Lake Basin. The maximum sorption capacity of 2,3,7,8-TCDT on sediment is under acidic conditions (pH < 4.0). Meanwhile, with the increase of DOM contents, the sorption of 2,3,7,8-TCDT in sediment was first promoted and then inhibited. In addition, the values of log*K*_oc_ were obtained and do not show a significant correlation with OC contents, which indicated that the partition effect was the dominating influencing factor of the sorption of 2,3,7,8-TCDT both in sediments and soil. Considering that many other factors could influence the sorption of 2,3,7,8-TCDT in the real environment, field experiments should be conducted. Meanwhile, the sorption behavior and mechanisms of other PCDT congeners in the aquatic environment should also be investigated in future works.

## Figures and Tables

**Figure 1 ijerph-19-11346-f001:**
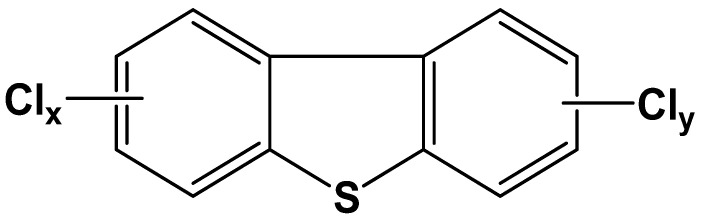
The structural formulas of PCDTs.

**Figure 2 ijerph-19-11346-f002:**
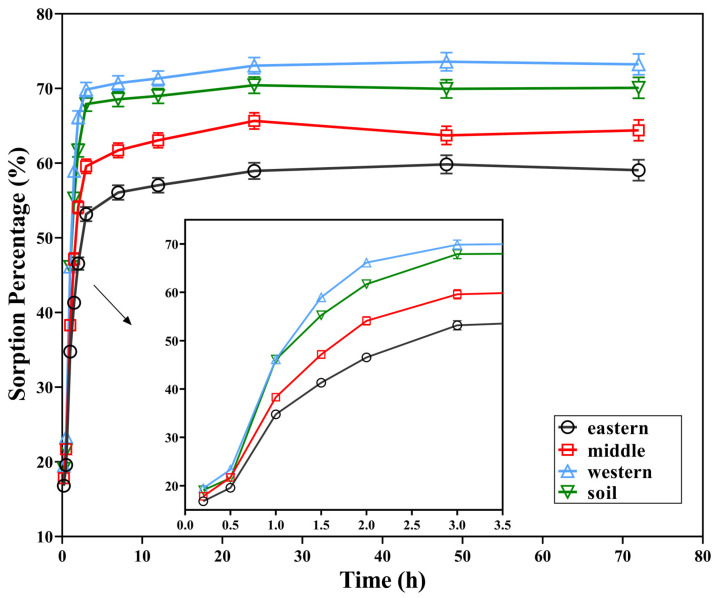
Sorption curves of 2,3,7,8-TCDT in three sediments from Chaohu Lake and paddy soil.

**Figure 3 ijerph-19-11346-f003:**
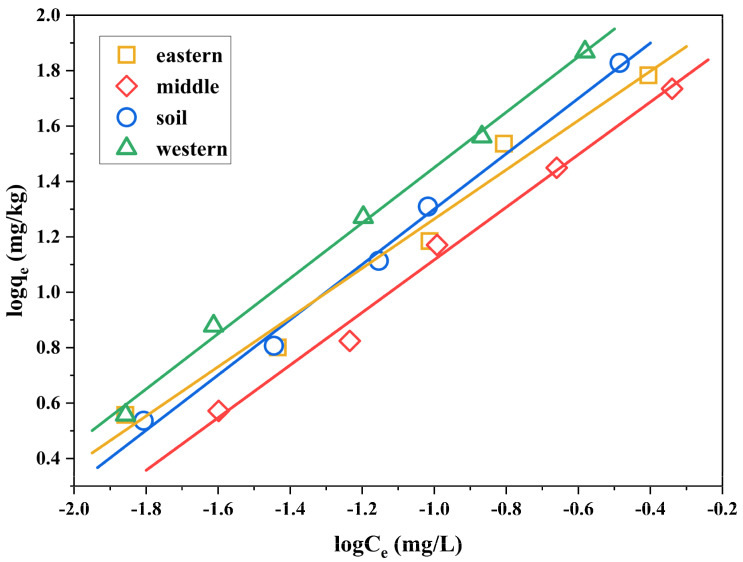
Freundlich sorption isotherms of 2,3,7,8-TCDT in three sediments from Chaohu Lake and paddy soil.

**Figure 4 ijerph-19-11346-f004:**
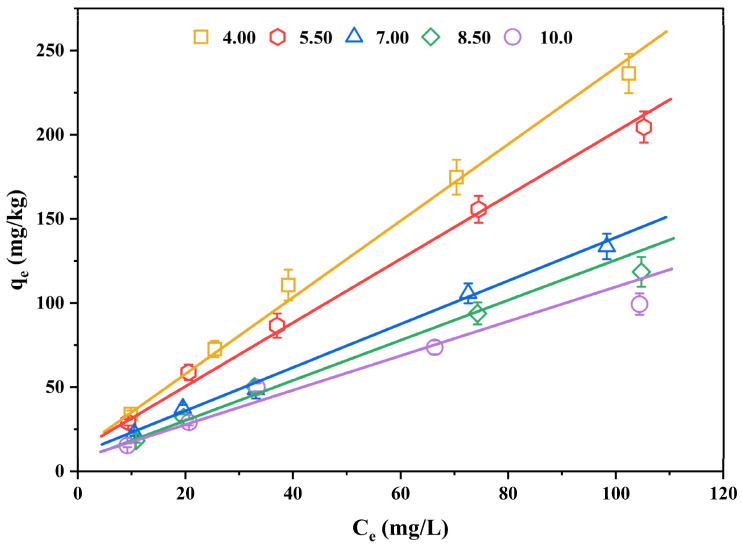
The isothermal sorption curves of 2,3,7,8-TCDT in sediments of the eastern Chaohu Lake at different pH values.

**Figure 5 ijerph-19-11346-f005:**
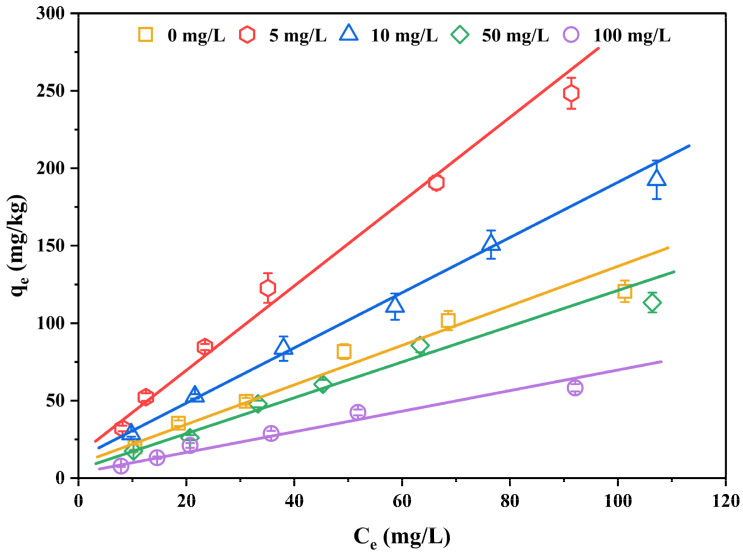
Isothermal sorption curves of 2,3,7,8-TCDT in the sediments of eastern Chaohu Lake under different DOM concentrations.

**Table 1 ijerph-19-11346-t001:** Physicochemical properties of sediment and soil samples.

Sample	Location	pH	Soil Texture			OC (%)	TN (%)
			Sand (%)	Silt (%)	Clay (%)		
Western part of Chaohu Lake	31°38’4″ N, 117°20’28″ E	7.07 ± 0.01	28.7 ± 0.10	61.7 ± 0.15	9.70 ± 0.03	3.59 ± 0.01	0.170 ± 0.02
Middle of Chaohu Lake	31°33’56″ N, 117°30’57″ E	7.03 ± 0.01	22.8 ± 0.09	65.7 ± 0.16	11.4 ± 0.04	3.49 ± 0.01	0.150 ± 0.01
Eastern part of Chaohu Lake	31°33’59″ N, 117°40’26″ E	7.63 ± 0.02	23.0 ± 0.09	67.0 ± 0.16	11.0 ± 0.04	4.45 ± 0.01	0.200 ± 0.02
Paddy soil sample	31°29’22″ N, 117°14’26″ E	5.70 ± 0.01	28.4 ± 0.10	60.0 ± 0.15	11.6 ± 0.04	4.55 ± 0.01	0.170 ± 0.02

**Table 2 ijerph-19-11346-t002:** The basic properties of the prepared straw DOM.

pH	TOC (mg/L)	Elemental Composition (%)			
		C	H	N	O
7.82 ± 0.07	295.5 ± 19.4	13.15 ± 0.08	2.46 ± 0.01	1.03 ± 0.01	12.20 ± 0.08

**Table 3 ijerph-19-11346-t003:** Freundlich model parameters of 2,3,7,8-TCDT in sediments and paddy soil.

Location	*K* _f_	*n*	*R* ^2^	*K* _f-oc_
Paddy soil sample	199 ± 8.43	1.0015 ± 0.01	0.994	43.8 ± 1.45
Western part of Chaohu Lake	282 ± 8.96	1.0004 ± 0.01	0.997	78.5 ± 2.08
Middle of Chaohu Lake	117 ± 5.78	1.0528 ± 0.01	0.991	33.5 ± 0.72
Eastern part of Chaohu Lake	143 ± 7.54	1.1246 ± 0.01	0.977	32.0 ± 0.68

**Table 4 ijerph-19-11346-t004:** Freundlich model parameters of 2,3,7,8-TCDT in sediments from the eastern part of Chaohu Lake at different pH values.

pH	*K* _f_	*n*	*R* ^2^
4.00	5.05 ± 0.06	1.20 ± 0.01	0.999
5.50	5.01 ± 0.06	1.25 ± 0.02	0.998
7.00	3.43 ± 0.03	1.26 ± 0.02	0.991
8.50	2.69 ± 0.02	1.22 ± 0.01	0.997
10.0	2.86 ± 0.02	1.29 ± 0.02	0.988

**Table 5 ijerph-19-11346-t005:** Freundlich model parameters of 2,3,7,8-TCDT in in sediments of eastern Chaohu Lake at different DOM levels.

DOM (mg/L)	*K* _f_	*n*	*R* ^2^
0	3.18 ± 0.03	1.24 ± 0.02	0.988
5	6.12 ± 0.06	1.21 ± 0.01	0.995
10	4.55 ± 0.03	1.25 ± 0.02	0.998
50	2.33 ± 0.02	1.18 ± 0.01	0.984
100	1.49 ± 0.02	1.20 ± 0.01	0.987

**Table 6 ijerph-19-11346-t006:** The *K*_d_ and log*K*_oc_ values of 2,3,7,8-TCDT in sediments from Chaohu Lake and paddy soil.

Location	*K* _d_	*f*_oc_ (%)	*K* _oc_	log*K*_oc_
Paddy soil sample	201 ± 8.56	4.55 ± 0.01	4410 ± 12.01	3.64 ± 0.03
Western part of Chaohu Lake	283 ± 6.55	3.59 ± 0.01	7877 ± 15.34	3.90 ± 0.04
Middle of Chaohu Lake	131 ± 4.60	3.49 ± 0.01	3760 ± 10.23	3.58 ± 0.02
Eastern part of Chaohu Lake	193 ± 6.87	4.45 ± 0.01	4338 ± 11.98	3.64 ± 0.03

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
