# Peer review of "An Insight into the Sorption Behavior of 2,3,7,8-Tetrachlorodibenzothiophene on the Sediments and Paddy Soil from Chaohu Lake Basin"

_ijerph, 2022, doi:10.3390/ijerph191811346_

Round 1

Reviewer 1 Report

In this manuscript, the authors report about the sorption behavior of a tetrachlorinated dibenzothiophene, a potentially toxic dioxin-like compound. Sorption was investigated at selected sediments and soil considering varying conditions like pH and dissolved organic matter. The work was carried out systematically and thoroughly, the conclusions are comprehensible and supported by literature. I recommend publishing this work after consideration of a few small formal suggestions for change.

1)      Please write out all abbreviations the first time they appear, e.g. Abstract, line 22: DOM; Main text: line 33: PCDF (here, polychlorinated dibenzothiophenes is duplicated); line 41: PCDD/DF; line 42: TCDT (it was written out in the abstract, but is needed another time in the main text); line 43: TCDD; line 58: PCDE; line 244: PCDPS; line 374: CDPS.

2)      Please provide errors in all tables like it is done in Table 1 for OC content.

3)      Figures 2 and 3: Please describe the samples more detailed (e.g. in the Figure caption).

4)      Lines 270 to 272: The last two sentences are redundant and can be omitted. The contents appear already at lines 255 to 256 and line 258 resp.

5)      Line 275: Please replace “greater” by “higher”; the same in line 277, replace “greatest” by “highest”.

6)      Line 285: The assignment of the values to the samples is incomplete.

7)      Chapter 3.5.: log values are in principal given without units.

Reviewer 2 Report

Review Report on the Manuscript Number: ijerph-1847468 Title: An insight into the sorption behaviour of 2,3,7,8-chlorinated dibenzothiophene on the sediments and paddy soil from Chaohu Lake basin

The manuscript investigates the sorption kinetics and isotherms of 2,3,7,8-tetrachlorined dibenzothiophene via batch equilibrium experiments onto three sediments and paddy soil from Chaohu Lake. I think that this manuscript must be revised before of be considered for publication. In the following, I suggest some possible improvements.

1.      I recommend summarize more accurate and representative keywords. Like the Sorption behaviour, Sediments and Paddy soil, just general keywords which are common, are they appropriate to be key words for this paper?

2.      The Abstract is very confusing for readers. The methods used in this study should be presented together.

3.      The content of introduction is confusing and very short, so it is suggested to further organize and refine.

4.      I suggest to add new references to introduction section (2020 or 2021).

5.      Line 54, Figure 1 is not needed for this section. Please remove it.

6.      Line 68, The Authors should report more about the previous results obtained for the study area.

7.      Line 97, You mentioned soil samples were collected from ecological paddy fields. It would be better to add pictures for each paddy field (in small scale).

8.      Line 118-119, The 0.01 M CaCl2 were used as the aqueous solvent phase to improve centrifugation and minimize cation exchange, and 0.05% HgCl2 was added as a biocide. What is the scientific basis for this setting?

9.      I suggest to separate results and Discussion sections.

10.  Line 219 to 222, here you present interesting results, you could explore and discuss these results further.

11.  Line 290-294, It would be better if you could report more results from other studies so that your readers can see how similar the findings were if the experimental conditions were comparable to your study.

12.  I am not really convinced of the quality of Figures shown in paper.

Reviewer 3 Report

Dear Editor, thank you very much for your invitation to review this manuscript ID: ijerph-1847468, submitted to the International Journal of Environmental Research and Public Health.

The original paper “An insight into the sorption behaviour of 2,3,7,8-chlorinated dibenzothiophene on the sediments and paddy soil from Chaohu Lake basin” by Nian et al. provides valuable information about the sorption behaviour of 2,3,7,8-chlorinated dibenzothiophene on sediments and soil and assess its potential environmental risk. Thus, the manuscript has merit for publication, however, the authors need to make a few improvements:

- English language: a minor spell check is required (a few misspellings). Please, review it.

- 2,3,7,8-chlorinated dibenzothiophene or 2,3,7,8-tetrachlorined dibenzothiophene?

- An excellent abstract needs to describe: The context, gap, purpose, methodology, results, and conclusions. I did not find the context and gap well described in the current abstract section, and the authors should consider including this information.

- Add “dissolved organic matter (DOM)” to the abstract.

Materials and Methods: I strongly recommend the authors provide more information in the subsection “2.2. Sample collection and characterization”. There are a few gaps to solve.

- “The sediment samples were collected from three parts of Chaohu Lake and the soil samples were collected from ecological paddy fields (Lujiang, Hefei, Anhui)”. Why did the authors collect just three points? Why were these points selected? What is the distance between these points? How many replicates were included in this study?

- Unfortunately, the quality of the figures is not high, and the authors should enhance their quality.

- “2.2. Data analyses”. The outcomes are presented as mean ± standard error, right? Provide this information.

Results and Discussion:
- The discussion is well supported.

Conclusion:
- Summarized. No comments.

References:
- References have been reformatted based on the journal’s suggested format. Please, review it.

Round 2

Reviewer 2 Report

I checked all answers and the authors did respond to all queries. I thus suggest to accept the manuscript.

With best regards

Reviewer 3 Report

Thank you for covering all my questions. Accept in current form.